# Model states for a class of chiral topological order interfaces

V. Crépel [1], N. Claussen[1], B. Estienne[2] & N. Regnault[1]

Interfaces between topologically distinct phases of matter reveal a remarkably rich phenomenology. To go beyond effective field theories, we study the prototypical example of such an interface between two Abelian states, namely the Laughlin and Halperin states. Using matrix product states, we propose a family of model wavefunctions for the whole system including both bulks and the interface. We show through extensive numerical studies that it unveils both the universal properties of the system, such as the central charge of the gapless interface mode and its microscopic features. It also captures the low energy physics of experimentally relevant Hamiltonians. Our approach can be generalized to other phases described by tensor networks.

[1] Laboratoire de Physique de l'École Normale supérieure, ENS, Université PSL, CNRS, Sorbonne Université, Sorbonne Paris Cité, Université Paris Diderot, 75005 Paris, France. [2] Laboratoire de Physique Théorique et Hautes Énergies, LPTHE, CNRS, Sorbonne Université, 75005 Paris, France. Correspondence and requests for materials should be addressed to V.C. (email: crepel@lpa.ens.fr)

The Integer Quantum Hall (IQH) effect, characterized by the quantization of the Hall conductance in units of $e^2/\hbar$, can occur even in the absence of a uniform magnetic field and requires neither flat bands nor Landau levels[1]. From a band-theory point of view, the difference between a trivial band insulator and the IQH phase stems from the subtle sensitivity to boundary conditions. This is encoded in the first Chern integer $\mathcal{C}$[2], which depends on the topology of the valence band bundle. This quantized invariant forbids an IQH phase to be adiabatically transformed into any trivial band insulator without undergoing a quantum phase transition[3]. Such a transition occurs for instance at the edge of an IQH droplet, which is nothing but an interface with the vacuum. The closure of the gap is manifest in the presence of conducting channels at the edge, and the corresponding Hall conductance $\mathcal{C}e^2/\hbar$ yields a measure of the Chern number $\mathcal{C}$. Strongly correlated phases with intrinsic topological order such as the Fractional Quantum Hall (FQH) effect share the same features: bulk global invariants control the nature of the possible gapless interface excitations at the transition to a trivial phase. Conversely, the Bulk-Edge Correspondence conjectures that the gapless theory at such a transition governs the full topological content of the FQH state (see for instance refs. [4–7] for in-depth studies and discussions about the validity of this correspondence). This conjecture was considerably substantiated by the pioneering work of Moore and Read[8] who expressed a large class of FQH model Wave Functions (WFs) as Conformal Field Theory (CFT) correlators. This CFT is chosen to match the one used to describe the gapless edge modes of the target state, making the correspondence between the bulk and edge properties transparent. It provides many insights into the study of these strongly correlated phases[9–13].

At the interface between two phases with distinct intrinsic topological order, the critical theory has no reason to be described by the same formalism. Indeed, only the mismatch in topological content of the two bulks is probed at such a gapless interface. For instance, the Hall conductance at the interface between two IQH plateaus is $\Delta\mathcal{C}e^2/\hbar$, controlled by the difference in Chern numbers $\Delta\mathcal{C}$. The notion of difference should be refined for strongly interacting systems such as FQH states, but even for the IQH, it cannot completely characterize the two bulks. As a consequence, even the interfaces between Abelian states have not yet been classified[14–16]. However, for interfaces with the same number of left and right movers, the topologically distinct ways of gapping the boundary were mathematically distinguished as Lagrangian subgroups[17,18].

Theoretical approaches to understand these interfaces mostly rely on the cut and glue approach[7], i.e., restricting the analysis to coupled one-dimensional modes. In this effective picture, both phases are solely described by their respective edge theories, and the interface emerges from the coupling between the two edge theories[19,20]. On the CFT side, this approach refines the notion of difference in topological content with the help of coset constructions[21–23]. In this article, we aim to put the above effective edge approaches on firmer ground by analyzing the full two-dimensional problem for a certain class of topological interfaces. To exemplify our method, we introduce model WFs describing the interface between two Abelian states, the Laughlin[24] and Halperin[25] states. Having a description of the full two-dimensional system, we can unveil both the universal properties and the microscopic features. We provide extensive numerical studies to probe the full critical theory at the interface, and to demonstrate that these WFs correctly capture the low energy physics of full two-dimensional system (see also ref. [26]). We discuss possible experimental realizations and the applicability of our method to other chiral topological order interfaces.

The Halperin ($m$, $m$, $m - 1$) state with $m$ integer (even for bosons and odd for fermions) appears at a filling factor $\nu = \frac{2}{2m-1}$. It describes a FQH fluid with an internal two-level degree of freedom[25,27] such as spin, valley degeneracy in graphene or layer index in bilayer systems. In the following, we use the terminology of the spin degree of freedom irrespectively of the actual physical origin. Such a state is the natural spin singlet[28–30] generalization of the celebrated, spin polarized, Laughlin state[24,31]. The latter describes an FQH state at filling factor $\nu = 1/m$. From now on, we will focus on bosons and $m = 2$, as this already realizes all the non-trivial physics we put forward in this article. Both the Halperin (221) and the Laughlin 1/2 states are the densest zero energy states of the following Hamiltonian projected onto the Lowest Landau Level (LLL)[32,33]:

$$\mathcal{H}_{\text{int}} = \int d^2\mathbf{r} \sum_{\sigma,\sigma'=\uparrow,\downarrow} : \rho_\sigma(\mathbf{r})\rho_{\sigma'}(\mathbf{r}) : + \mu_\uparrow\rho_\uparrow(\mathbf{r}), \qquad (1)$$

respectively, for $\mu_\uparrow = \infty$ and $\mu_\uparrow = 0$. Here $\mu_\uparrow$ is a chemical potential for the particles with a spin up, $\rho_\sigma$ denotes the density of particles with spin component $\sigma$, and :.: stands for normal ordering. Hence, creating an interface between these two topologically ordered phases can be achieved by making $\mu_\uparrow$ spatially dependent without tuning the interaction[21].

The Laughlin and Halperin model WFs can also be defined by CFT correlators[8,34]. Both states are Abelian, and the primaries appearing in the correlators are vertex operators[13,35–38]. The mode expansion of the primaries in the correlators allow for an exact MPS description of the WFs over the Landau orbitals[39–41]. On the cylinder geometry, Landau orbitals are shifted copies of a Gaussian envelope whose center is determined by the momentum along the compact dimension (see Fig. 1). Because of this translation symmetry on the cylinder geometry, the exact MPS representation of the model WFs can be made site independent which enables the use of efficient infinite-MPS (iMPS) algorithms[42,43]. The auxiliary space, i.e., the vector space associated to the matrix indices, is the truncated CFT Hilbert space $\text{Aux}_H$ used for the Halperin state, a compactified two-component boson[44]. Similarly, the CFT Hilbert space for the Laughlin state $\text{Aux}_L$ is made of a single compactified boson. By choosing the same $m$ for the two model WFs, it is possible to embed the auxiliary space of the Laughlin state in that of the Halperin state. More precisely, up to compactification conditions[26,44] we have $\text{Aux}_H = \text{Aux}_L \otimes \text{Aux}_\perp$ where $\text{Aux}_\perp$ is the CFT Hilbert space for a free boson $\varphi_\perp$. Representing the vertex operators in the same product basis for $\text{Aux}_H$ leads to the Halperin and $B_H^{(n^\downarrow, n^\uparrow)}$ and Laughlin ($B_L^{(n^\downarrow)} \otimes \mathbb{I}$) iMPS matrices[44] (here the physical indices $n^\uparrow$ and $n^\downarrow$ denote the orbital occupation for particles with a spin $\uparrow$ and $\downarrow$).

This embedding is the key to our approach to the interface between two distinct topological orders. Indeed, the canonical choice of basis for the CFT Hilbert space of a free boson[35,41,44] allows to identify all auxiliary indices at the transition such that the gluing of the two phases becomes a mere matrix multiplication in the MPS language. Similarly, we can keep track of the quantum numbers at the interface and hence use the block structure of the iMPS matrices with respect to the U(1)-charges and momentum of the bosons. This provides an additional refinement and enhances the efficiency of the iMPS machinery[39,42,43]. Finally, the truncation of the auxiliary space is constrained by the entanglement area law[45], the bond dimension should grow exponentially with the cylinder perimeter $L$ to accurately describes the model WFs.

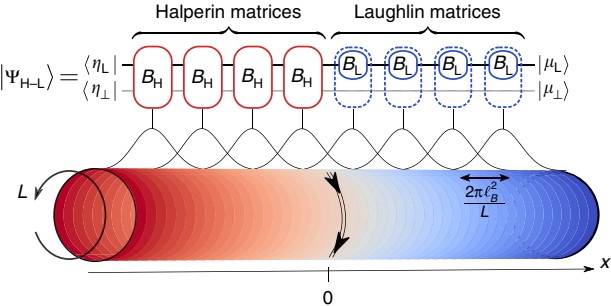

**Fig. 1** Model Wavefunction for the Interface: Schematic representation of the MPS ansatz $|\Psi_{H-L}\rangle$ for the Halperin 221–Laughlin 1/2 interface on a cylinder of perimeter $L$ (i.e., we assume periodic boundary condition along the y-axis). The momentum along the compact direction labels the Landau orbitals and determines the center of their Gaussian envelope on the cylinder (schematically depicted on top of the cylinder). The Halperin iMPS matrices $B_H$ (red) are glued to the Laughlin iMPS matrices $B_L \otimes \mathbb{I}$ (blue) in the Landau orbital space. Due to the embedding of one auxiliary space into the other, the quantum numbers of $|\mu_\perp\rangle$ (see Eq. (2)) are left unchanged by the Laughlin matrices all the way to the interface. It constitutes a direct access controlling the states of the interface chiral gapless mode, graphically sketched here with a double arrow

## Results

**Model wavefunction**. Our ansatz may be understood as an abrupt change of the chemical potential $\mu_\uparrow$ from zero to infinity in orbital space, which maps to a smooth Gaussian ramp in real space over a distance $2\pi \ell_B^2/L$, $\ell_B$ being the magnetic length. Polarizing the system in its spin down component amounts to using the Laughlin iMPS matrices for any Landau orbitals in the polarized region $x \geq 0$, while Halperin iMPS are used when $x < 0$. Thus our MPS, schematically depicted in Fig. 1, reads

$$\langle \{n_k^\downarrow\}_{k\in\mathbb{Z}}, \{n_k^\uparrow\}_{k<0} | \Psi_{H-L}\rangle = \underbrace{\langle \eta_L| \otimes \langle \eta_\perp|}_{\langle \eta|} \cdots B_H^{(n_{-2}^\downarrow, n_{-2}^\uparrow)} B_H^{(n_{-1}^\downarrow, n_{-1}^\uparrow)}$$
$$(B_L^{(n_0^\downarrow)} \otimes \mathbb{I})(B_L^{(n_1^\downarrow)} \otimes \mathbb{I}) \cdots \underbrace{|\mu_L\rangle \otimes |\mu_\perp\rangle}_{|\mu\rangle}, \quad (2)$$

where $|\{n_k^\downarrow\}_{k\in\mathbb{Z}}, \{n_k^\uparrow\}_{k<0}\rangle$ is the many-body orbital occupation basis. Here $\langle\eta|$ and $|\mu\rangle$ are the two states in the auxiliary space fixing the left and right boundary conditions. Due to its block structure, our MPS naturally separates the different charge and momentum sectors all along the cylinder[44]. In particular, it distinguishes the different topological sectors through the U(1)-charges of the MPS boundary states of Eq. (2). While for the Halperin state, the U(1)-charge of both $\langle\eta_L|$ and $\langle\eta_\perp|$ should be fixed to determine the topological sector, only the one of $|\mu_L\rangle$ is required to fix the topological sector of the Laughlin phase. The remaining bosonic degree of freedom, $|\mu_\perp\rangle$ at the edge of the Laughlin bulk constitutes a knob to dial the low-lying excitations of the one-dimensional edge mode at the interface (remember that due to the specific product-basis choice, the Laughlin iMPS matrices act as the identity on $|\mu_\perp\rangle$ and propagate the state all the way to the interface—see Fig. 1).

A direct probe of the interface is the spin resolved densities $\rho_\uparrow$ and $\rho_\downarrow$ of the model state presented in Fig. 2a for the transition between the Halperin (221) and the Laughlin 1/2 states. They smoothly interpolate between the polarized Laughlin bulk at filling factor $\nu_L = 1/2$ and the Halperin unpolarized bulk at filling factor $\nu_H = \frac{1}{3} + \frac{1}{3}$. We recover the typical bulk densities and the spin SU(2) symmetry of the Halperin (221) state after a few magnetic lengths, which is much larger than the distance between

orbitals. This is not an artefact of our ansatz since Exact Diagonalization (ED) simulations of Eq. (1) for a half polarized system with delta interactions show the same behavior. The density inhomogeneity at the interface is a probe of the interface reconstruction due to interactions (see Eq. (1)).

**Numerical results**. A crucial feature that our ansatz should reproduce is the topological order of the Halperin and Laughlin bulks away from the interface, i.e., when $|x| \gg \ell_B$. Local operators such as the density cannot probe the topological content of the bulks. We thus rely on the entanglement entropy (for a review, see ref. [46]) to analyze the topological features of our model WF. Consider a bipartition $\mathcal{A} - \mathcal{B}$ of the system defined by a cut perpendicular to the cylinder axis at a position $x$. The Real-Space Entanglement Spectrum (RSES)[47–49] and the corresponding Von Neumann EE $S_\mathcal{A}(L, x)$ are computed for various cylinder perimeters $L$ using techniques developed in refs. [39,41,44]. Two-dimensional topological ordered phases satisfy the area law[45]

$$S_\mathcal{A}(L, x) = \alpha(x)L - \gamma(x), \quad (3)$$

where $\alpha(x)$ is a non universal constant and $\gamma(x)$ is the Topological Entanglement Entropy (TEE). The latter is known to characterize the topological order[50,51]. Since the cylinder perimeter is a continuous parameter in our simulations, we extract these constants by numerically computing the derivative $\partial_L S_\mathcal{A}(L, x)$ as depicted in Fig. 2b for $\gamma(x)$. Deep in the bulks, our results match the theoretical prediction for the Laughlin $(\gamma(x \to +\infty) = \log\sqrt{m})$ and Halperin $(\gamma(x \to -\infty) = \log\sqrt{2m-1})$ states. This proves the validity of our ansatz away from the interface.

Near the interface, the EE still follows Eq. (3) as was recently predicted for such a rotationally invariant bipartition[19]. The correction $\gamma(x)$ smoothly interpolates between its respective Laughlin and Halperin bulk values (see Fig. 2b). Hence, it contains no universal signature of the critical mode at the interface between the two topologically ordered phases. The same conclusion holds for the area law coefficient $\alpha(x)$ (see Supplementary Fig. 3).

We now focus on the critical mode that should lie at the interface. Effective one-dimensional theories similar to the ones of refs. [21,23] predict that the gapless interface is described by the free bosonic CFT $\varphi_\perp$, of central charge $c = 1$ and compactification radius $R_\perp = \sqrt{m(2m-1)}$, which is neither an edge mode of the Halperin state nor of the Laughlin state. It may be understood as follows: the edge of the Halperin FQH droplet is a spinful Luttinger liquid in which spin and charge excitations separate into two independent bosonic excitations denoted as $\varphi_c$ and $\varphi_s$. Because the interface presented is fully transmissive to spin down bosons, backscattering processes[52] gap out the combination of spin and charge bosons relative to spin down particles. What remains is the non-trivial bosonic field

$$\varphi_\perp = \sqrt{\frac{1}{2m}}\varphi_c + \sqrt{\frac{2m-1}{2m}}\varphi_s. \quad (4)$$

To test this one-dimensional effective theory, we compute the RSES for a bipartition for which the part $\mathcal{A}$ consists of a rectangular patch of length $\ell$ along the compact dimension and width $w$ along the x-axis, i.e., we break the rotational symmetry along the cylinder perimeter. To fully harness the power of the iMPS approach, it is convenient to add a half infinite cylinder to the rectangular patch (see Fig. 3a). The technical challenges inherent to such a computation breaking spatial symmetries are presented in ref. [26]. We isolate the contribution of the interface edge mode from the area laws and corner contributions with a Levin–Wen addition subtraction scheme[51] depicted in Fig. 3a. Note that it counts the contribution of the gapless interface to the

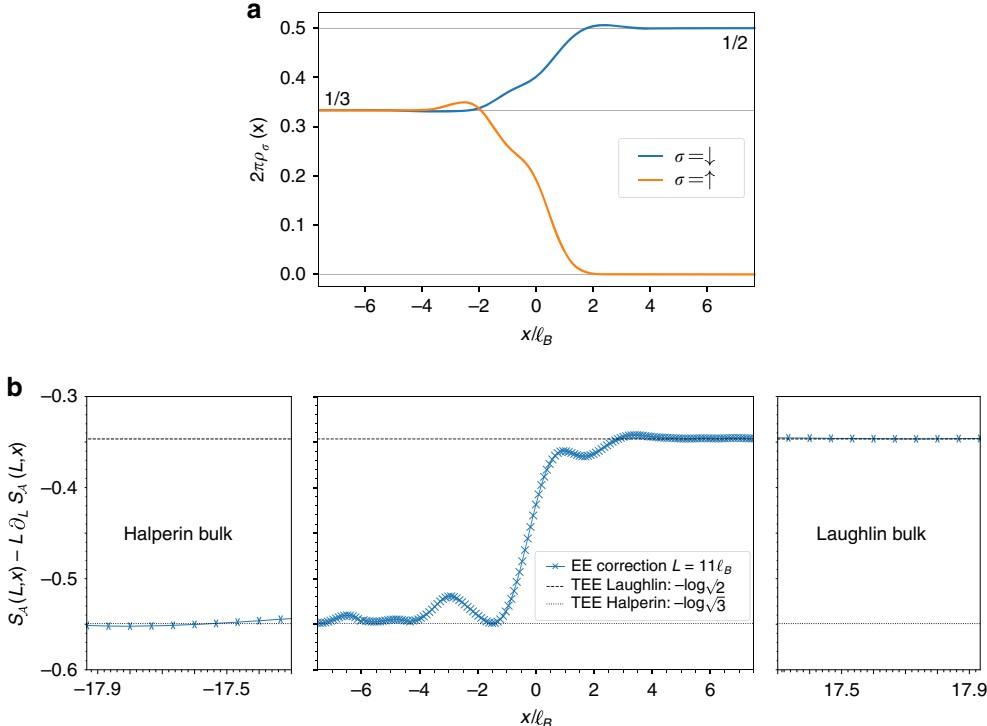

**Fig. 2** An interface between two toplogical orders: **a** Spin resolved densities of the MPS ansatz state along the cylinder axis. They smoothly interpolate between the Laughlin ($2\pi\rho_\downarrow = 1/2$ and $\rho_\uparrow = 0$) and the Halperin ($2\pi\rho_\downarrow = 2\pi\rho_\uparrow = 1/3$) theoretical values. The persistent density inhomogeneity is an edge reconstruction due to interactions (see Eq. (1)). **b** Constant correction of the EE to the area law for a rotationally symmetric bipartition at position $x$. The correction is found to be constant (see Eq. (3)). The extraction of the TEE deep in the Laughlin and Halperin bulks shows that the MPS ansatz correctly captures the topological properties away from the interface

EE twice and that a spurious contribution due to short range entanglement along the cut parallel to the $x$-axis may appear. We vary the length $\ell$ along the compact dimension of the cylinder while keeping $w$ constant. In Fig. 3b, we observe that the EE $S_{\mathcal{A}}(\ell, w)$ resulting from the Levin–Wen scheme shown in Fig. 3a follows the 1D prediction for a chiral $c = 1$ CFT with periodic boundary conditions[53]

$$S_{\mathcal{A}}(\ell, w) - S_{\mathcal{A}}(L/2, w) = 2\frac{c}{6}\log\left[\sin\left(\frac{\pi\ell}{L}\right)\right]. \qquad (5)$$

To be more quantitative, we fit the numerical derivative $\partial_\ell S_{\mathcal{A}}(\ell, w)$ with the theoretical prediction using the central charge $c$ as the only fitting parameter (the derivative removes the area law contribution arising from the cut along $x$). We minimize finite size effects by keeping only the points for which $\ell$ and $L - \ell$ are both greater than four times the Halperin bulk correlation length[44]. Fitting the data obtained for a perimeter $L = 12\ell_B$ (the largest $L$ that reliably converges with the the largest reachable auxiliary space), we find $c \simeq 0.987(1)$ (see Fig. 3). We verified that the rectangular patch was covering entirely the gapless mode by checking that the results hold for a large range of $w$ (see Supplementary Fig. 4). Moreover, performing the same calculation far away in the gapped Laughlin phase leads to a fitted value of $c \leq 0.13$ (see Methods).

In order to fully characterize the gapless mode circulating at the interface, we now extract the charges of its elementary excitations which are related to the compactification radius $R_\perp = \sqrt{6}$. As previously mentioned, excited states of the critical theory are numerically controlled by the U(1)-charge $N_\perp$ which is part of the MPS boundary condition on the Laughlin side. For each of these excited states, we compute the spin resolved

densities and observe that the excess of charge and spin are localized around the interface (see inset of Fig. 4). They stem from the gapless interface mode observed in Fig. 3b and we plot the charge and spin excess as a function of $N_\perp$ in Fig. 4. The linear relation indicates that the interface critical theory hosts excitations carrying a fractional charge $e/6$. Physically[21], a Laughlin 1/2 quasihole carrying an electric charge $e/2$ passing through the transition region can excite an elementary Halperin 221 quasihole of charge $e/3$ but $e/6$ charge has to be absorbed by the gapless mode at the interface.

## Discussion

While our ansatz has the desired features to describe the low energy physics of Eq. (1), we can provide a more quantitative comparison. As opposed to a Halperin state with a macroscopic number of quasiholes in the polarized region (as discussed in ref. [21]) which completely screens the Hamiltonian Eq. (1), the critical edge mode at the interface now acquires a finite energy. But this low energy mode clearly detaches from the continuum, allowing us to compare it to our ansatz in finite size studies using exact diagonalization. The largest accessible system size involves 13 bosons, 9 with spin down, and 4 with spin up, interacting with delta potentials over 21 orbitals, 9 of which are completely polarized. We find extremely good agreement between the ED ground state and our MPS ansatz in finite size with vacuum boundary conditions at the edge, with an overlap of 0.9989. This provides a clear evidence of its physical relevance. To capture the low energy features above the ground state, we can change the interface gapless mode momentum by choosing the correct level descendant $P_\perp$ of the $\varphi_\perp$ boson on the Laughlin side. We are able to reproduce the first few low-lying excitations of the system above the finite size ground state with great accuracy[26]. Using matrix

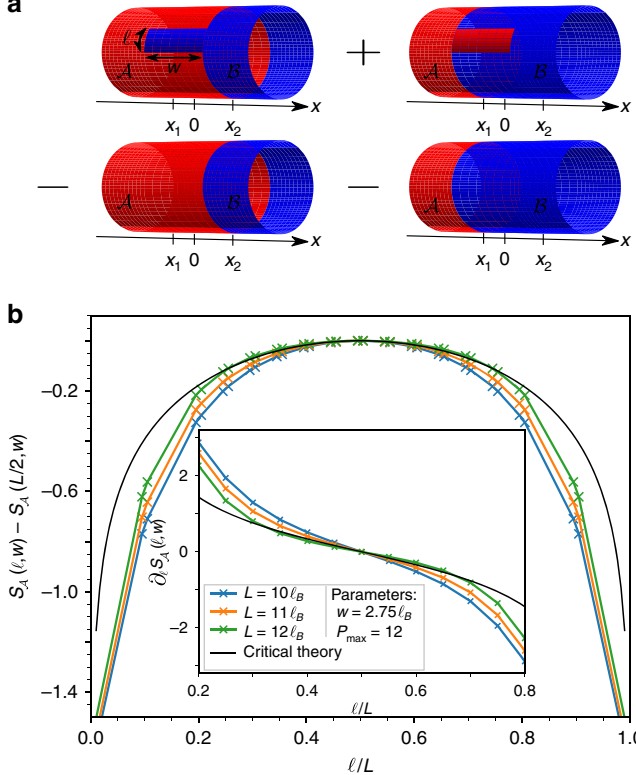

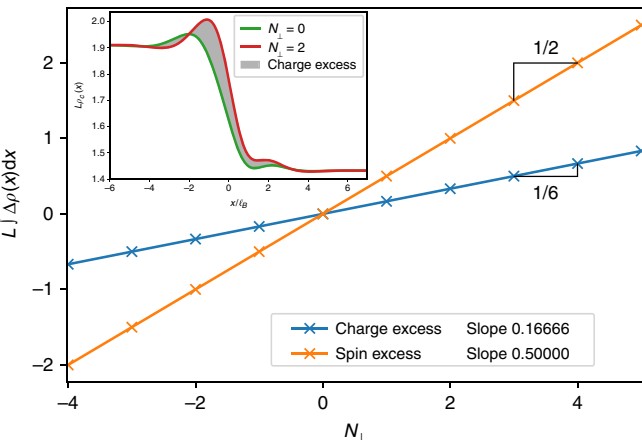

**Fig. 4** Spin and charge fractionalization: Charge and spin excess are localized at the interface when excited states are addressed via the MPS boundary U(1) charge $N_\perp$. Inset shows how to extract the charge excess (gray shaded area) from the charge densities $\rho_c$ at different $N_\perp$. Each excess follows a linear relation with extremely good accuracy. The charge (resp. spin) excess has a slope $0.166662(5) \simeq 1/6$ (resp. $0.500005(5) \simeq 1/2$). This indicates that the elementary excitations of the $c = 1$ critical theory (cf. Fig. 3) at the interface carry a fractional charge $e/6$ and a fractional spin $1/2$ in unit of the bosonic spin

**Fig. 3** Chiral interface edge mode: **a** Levin–Wen subtraction scheme to get rid of the spurious area law coming from the patch boundaries along the cylinder perimeter together with corner contributions to the EE. The position and width $w$ of the rectangular extension are selected to fully include the gapless mode at the interface. The critical EE coming from the gapless mode at the interface is counted twice. **b** $S_\mathcal{A}(\ell, w)$ for different cylinder perimeters. They all fall on top of the CFT prediction Eq. (5) with $c = 1$ (black line), pointing toward a chiral Luttinger liquid at the interface. The inset shows the derivative $\partial_\ell S_\mathcal{A}(\ell, w)$ and its agreement with the theoretical prediction

product operators and our ansatz, we can actually focus on and evaluate the dispersion relation of the gapless interface mode[26].

We also considered the fermionic interface between the Laughlin 1/3 and Halperin (332) states which is more relevant for condensed matter experiments[26]. It exhibits the same features as the bosonic case previously discussed: the gapless mode at the interface is described by a bosonic $c = 1$ CFT $\varphi_\perp$ whose elementary excitations agree with the value of $R_\perp = \sqrt{15}$. In the latter case, experimental realization of this interface can be envisioned in graphene. There, the valley degeneracy leads to a spin singlet state at $\nu = 2/5$[54,55] while the system at $\nu = 1/3$ is spontaneously valley polarized[55–57]. Thus, changing the density through a top gate provides a direct implementation of our setup.

Our model WF not only captures the bulk topological content of the FQH states glued together, but it also faithfully describes the gapless interface theory, as shown by the extensive numerical analysis presented in this article. This gapless $c = 1$ theory hosts fractional elementary excitations, which are neither Laughlin nor Halperin quasiholes, a probe of the interface reconstruction due to interactions. Although the universal properties can be inferred from one-dimensional effective theories[14,17,52], our ansatz validates such an approach while granting access to a full microscopic characterization. It also accurately matches the low energy physics of experimentally relevant microscopic Hamiltonians.

Our approach can be extended to topologically ordered phases described by MPS or any tensor network method, as long as there is an embedding of one auxiliary space into another. It paves the way to a deeper understanding of interfaces between such phases.

## Methods

**Levin–Wen subtraction scheme**. We come back to the RSES for a bipartition consisting of a rectangular patch of width $w = x_2 - x_1 > 0$ along the cylinder axis and a length $\ell \in [0, L]$ around the cylinder perimeter, $y_2 = -y_1 = \ell/2$. A half infinite cylinder is added to the patch (see Fig. 3a) in order to be able to switch from a site-dependent and weighted MPS to the iMPS matrices far from the transition. We first would like to qualitatively enumerate the possible contributions to the EE for such a cut. We see three possible contributions for the left topmost cut of Fig. 3a:

- Area law terms[45]

$$\alpha(x_1)\ell + \alpha(x_2)(L - \ell) + 2\int_{x_1}^{x_2} \alpha(u)\,du, \quad (6)$$

where the linear coefficient at position $x$ is denoted as $\alpha(x)$ (see Eq. (3)).
- Possible chiral critical mode contributions[53]

$$\frac{c}{6}\log\left[\sin\left(\frac{\pi\ell}{L}\right)\right], \quad (7)$$

where $c$ is the central charge.
- Other constant corrections to the area law or corner contributions.

The addition subtraction scheme described in the main text, which is nothing but a Levin–Wen type cut[51], removes the area law terms at $x_1$ and $x_2$ together with the corner contributions. Hence, up to a constant $f(w)$ which depends on $w$, we expect the resulting EE $S_\mathcal{A}(\ell, w)$ to have the following form:

$$S_\mathcal{A}(\ell, w) = 2\frac{c}{6}\log\left[\sin\left(\frac{\pi\ell}{L}\right)\right] + f(w). \quad (8)$$

By considering $S_\mathcal{A}(\ell, w) - S_\mathcal{A}(L/2, w)$, we get rid of $f(w)$ and we obtain Eq. (5) in the main text. We find a very good agreement with the numerical results, presented in Fig. 3. The numerical extraction of $S_\mathcal{A}(\ell, w)$ is detailed in ref. [26], and additional results on the convergence of the EE can be found in the Supplementary Note 2. Another way to get rid of the constant corrections or of any pure function of $w$ is to focus on the derivative $\partial_\ell S_\mathcal{A}(\ell, w)$. It allows a one-parameter fit of the numerical data on the theoretical prediction Eq. (5). We exemplify the fitting procedure described above in Fig. 5 where we clearly see that the critical contribution to $\partial_\ell S_\mathcal{A}(\ell, w)$ is only present when the patch covers the interface. Deep in either of the two bulks, we only expect no critical contributions to the EE and the extraction of the central charge indeed gives $c_\text{Fit} = 0.072$ (resp. $c_\text{Fit} = 0.127$) deep in the Halperin (resp. Laughlin) phase. This consistency check indicates that the features observed

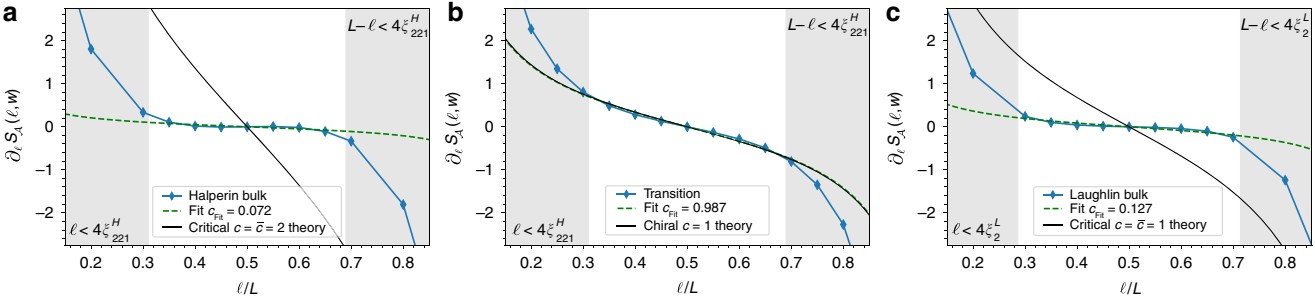

**Fig. 5** Critical contribution of the EE at the interface: fitting procedure to characterize the critical contribution to the EE $S_{\mathcal{A}}(\ell, w)$ for a Levin–Wen type cut depicted in Fig. 3. **a** far in the Halperin bulk, **b** for a patch covering the interface, and **c** deep in the Laughlin phase. All data points were taken for a cylinder perimeter $L = 12\ell_B$, a patch width $w = 2.75\ell_B$ and a truncation parameter $P_{max} = 12$. To mitigate finite size effects, we discard the points for which $\ell$ or $L - \ell$ are smaller than four times the correlation length (gray shaded area in the plots). To estimate these regions we use the bulk Halperin 221 correlation length $\xi_{221}^H \simeq 0.941\ell_B$ in **a**, **b** and the Laughlin 1/2 bulk correlation length $\xi_2^L \simeq 0.858\ell_B$, which we had previously extracted[44]. We only see a critical contribution when the patch covers the transition region and is large enough (see Supplementary Fig. 4), as expected. The prediction from critical theories are plotted in black while the result of the fit are shown with dashed green lines

at the interface are not a mere artifact of our numerical analysis but the actual signature of a critical mode.

## Data availability

Raw data and additional results supporting the findings of this study are included in Supplementary Notes and are available from the corresponding author on request. The exact diagonalization data for finite size comparisons have been generated using the software "DiagHam" (under the GPL license).

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

## Acknowledgements

We thank E. Fradkin, J. Dubail, A. Stern, and M.O. Goerbig for enlightening discussions. We are also grateful to B.A. Bernevig and P. Lecheminant for useful comments and collaboration on previous works. V.C., B.E., and N.R. were supported by the grant ANR TNSTRONG No. ANR-16-CE30-0025 and ANR TopO No. ANR-17-CE30-0013-01.

## Author contributions

V.C., N.C., and N.R. developed the numerical code. V.C., N.C., and N.R. performed the numerical calculations. V.C., N.C., B.E., and N.R. contributed to the analysis of the numerical data and to the writing of the manuscript.

## Additional information

**Competing interests:** The authors declare no competing interests.

