## [Peer Review File · Nature Communications]

Reviewers' comments:

Reviewer #1 (Remarks to the Author):

Dear Authors and Dear Editors of Nature Communications,

The paper "A variational approach to chiral topological order interfaces" represents an important advance in the description of systems which have spatial regions in different topologically ordered phases, with nontrivial low energy degrees of freedom on the edges between these regions. The paper introduces trial wave functions for an important class of examples of such systems which may be experimentally realized, for example in fractional quantum Hall systems and which are shown to give an excellent description of the exact ground states as well as some of the low energy excitations. In particular, the charge and spin of the edge excitations can be calculated from the trial wave functions, which do not contain input parameters that correspond to these quantities. The class of trial wave functions considered is versatile and can be used in future work to model Hamiltonians for a variety of more realistic experimental situations (typically with less sharp edges in the chemical potential). I think the work presented in the paper and in the supplementary data is excellent and recommend it for publication.

That said, I do have some remarks that the authors may wish to take into account.

First of all, reading the paper and supplementary information I felt that I was reading a long paper masquerading as a short paper. Much of the material that has been relegated to the supplementary information would have been completely appropriate for the paper (I expect it was pushed out by the page limit). For example the exact diagonalization study which confirms the validity of the trial wave functions is a significant work in itself. It establishes for example the charge and spin of edge excitations directly from the Hamiltonian, rather than from a trial wave function. The study of the fermionic 332/Laughlin 1/3 state (as opposed to the bosonic 221/Laughlin 1/2 state) is also relegated to the supplementary information. While it may seem repetitive to include this in the paper and the data may not look quite as good as for the 221 state, nevertheless the 332 state is more likely to be observed in experiments (certainly in the short term) and with a few more pages it would have been logical to include the relevant graphs in the paper. The authors also did a significant amount of work on the excited states of the system, establishing how some of these can be identified as excitations of a low lying edge mode, constructing trial wave functions for these and then checking their validity. This is only briefly mentioned in the paper itself.

On the other hand, there is quite a lot of space devoted in the paper to the confirmation that the trial wave functions produce the correct topological orders away from the edge, by calculation of the charge/spin densities and the real space entanglement entropy. While this is an important check and requires highly nontrivial calculation, it would be a great surprise if this failed, given the construction of the trial states.

In any event, the fact that there are 14 pages of supplement against 6 pages of paper (just over 4 excluding references) makes it clear that a lot is hidden from first view here. I would urge the authors to consider if this is the most appropriate form in which to present their findings.

I also have some minor remarks on the substance of the paper.

1.

On page three is mentioned that

"The remaining boson at the edge of the Laughlin bulk constitutes a knob to dial the states of the one-dimensional edge mode..."

I think it would be good to clarify a bit more what is meant here or otherwise to indicate where this is clarified.

2.

in FIG. 2 and various other figures (in the supplement), there are lines representing the expected results for the critical theory. It is not clear to me whether these represent the theoretical curves at $c=1$ or the fit which leads to the value $c=0.987(1)$ (mentioned on page 4), (Similarly later for the 332 state in the supplement). Perhaps these are indistinguishable by eye, but it would be good to mention (although I may have missed it).

3.

In the supplement, the Virasoro mode L_0 appears in a number of places, notably in formula (10). It could cause some confusion with the length L and it would be good to mention explicitly that the Virasoro mode is meant (as is done later when it reappears on page 4 of the supplement).

4.

In section II.B of the supplement, it is noted that the spectrum has low energy features that clearly detach from the continuum. This is evident in Fig. 5(b) which shows the spectrum. However that figure employs a log scale. In a linear plot, there would probably not be a very clear separation of energy scales. Could the authors comment on whether/why they expect a clear separation between edge modes and continuum in the thermodynamic limit?

Some less obvious typos etc.:

- In a few places in paper and supplement, "bound dimension" appears instead of "bond dimension"
- In the caption to Fig. 2 "get rid off" -> "get rid of"
- In the supplement, footnote/reference marks appear in confusing places a few times. Note for example the appearance of $|\det K|=3^4$ in the paragraph below formula (17), as well as $L_{\{0\}}^{\{6\}}$ further down that paragraph.
- supplement, section VII, paragraph 1: "Much larger than the later" should be "Much larger than the latter"

Reviewer #2 (Remarks to the Author):

In this work the authors explore a variational ansatz for the chiral modes at the boundary between two different fractional quantum Hall functions. The ansatz "glues" together the exact matrix-product-state ansatz for two FQH states. The authors then show this method can be used to verify a number of expected properties such as the charge, spin, and log-divergent entanglement of the edge mode.

I think this is a great paper, my main issue was one of clarity: from the text I could not precisely determine what the variational ansatz was, though I'm pretty certain I know what you refer to. You don't explain much beyond

"Our variational ansatz for the transition consists in using the Halperin iMPS matrices for any Landau orbital whose center is in the unpolarized region $x < 0$, and the Laughlin iMPS matrices when $x > 0$, as sketched in Fig. 1a. "

To make sure I understand, this is an ansatz with zero (continuous) variational parameters? More generally, you could insert a matrix acting on the ancilla space at the location where the MPS are glued together, and treat the matrix as a variational parameter. If so, when you glue them together what "gauge" of the MPS do you use? (since, in general, gluing is not a gauge-invariant prescription) Another way to put this: in the figure, you write "Halperin matrix" and "Laughlin matrix", but really

there are different way you can group things (a subset of gauge redundancy $A \rightarrow X A X^{-1}$) which redefines the matrix. I assume you mean you glue together the exact MPS before putting the MPS into a canonical form, e.g., you just suddenly switch which zero mode and vertex operator to include. I suppose the key here is that the $1/3$ ancillary space can be viewed as a "subset" of the $2/5$ ancillary space, for a more general boundary there wouldn't be an obvious prescription like this. And, to be clear, you then contract over all the indices in a given topological sector of the $1/3$ state?

Also, when you explore other topological sectors with N_{edge} , should it be obvious to me how you choose to glue together the momenta sectors? The center of mass of the charged excitation will depend on the total momenta sector - I suppose you are again making use of the special " $1/3$ subset $2/5$ " structure, and the simple factorization of the charge N and descendent index P_{edge} of the chiral boson?

As you can see, all of these issues would be clarified with some equations precisely stating the ansatz. Also, when you say "This ansatz may be understood as an abrupt change of the chemical potential ...", are you speaking figuratively? For example, I would doubt the orbital density of your ansatz is literally a step function.

In the end, if you are doing what I assume you are the calculation makes sense and should be published in Nature Comm.

Reviewer #3 (Remarks to the Author):

The work by Crepel et al, "A Variational Approach to Chiral Topological Order Interfaces" is an impressive piece of science. It puts into practice a technique for constructing a wavefunction interface which has long been discussed in the community as being possible, but has never previously been realized (at least not to my knowledge). Further, the resulting approach is thoroughly analyzed using a number of sophisticated techniques to fully establish that the resulting wavefunctions have properties as expected.

The overall scheme of the paper is to construct an interface between two different topological orders using a sharp interface in MPS space. The resulting wavefunction interface, now obtained on the microscopic level, is tested in quite a number of ways to probe its topological properties.

Generally think strikes me as very nice work which should be interesting to a fairly broad audience and would be entirely appropriate for Nature Communications. The work is extremely well developed and (including the supplementary material) is very well explained.

I have a number of comments which I list below. While some of my comments bring up interesting issues that the authors might consider discussing (if they have simple answers) I don't think any of my comments and suggestions are at all crucial --- the work could be published in the current form.

List of comments:

(1) Most importantly: I think the authors should discuss for what interfaces this technique will be possible. It seems one must be able to embed one auxiliary space in another. While there are several cases of topological interfaces where this occurs, it is certainly not generic. Admittedly in many cases if one tries to put together two topological orders, one simply gets the two edges of the two

constituents without anything interesting happening at the interface. However, I think there are cases where there nontrivial things happening at the interface, which still do not fit into the structure discussed in this work -- or at least not in any obvious way.

(2) Given (1) I think the title of the paper might be reconsidered. Maybe ". . . to CERTAIN chiral topological order interfaces" might be more appropriate.

(3) On the topic of the title of the paper --- while it is true that we often call the Laughlin state "variational", and so one could equally well call the current approach "variational", I think this misrepresents what they are (afterall, we don't really "vary" anything). "Trial Wavefunction" might be more accurate. Possibly an even more accurate title would be "MPS approach to certain chiral topological order interfaces"

(4) PLEASE: Fix the spelling of the word "subtraction" (not "substraction"). In the supplement too. (and in one place you have "bound" dimension, where I think you mean "bond" dimension).

(5) The authors in the intro use a few bits of language which are not fully precise.

Perhaps the most serious error is to say that the insulating bulk completely determines the gapless theory at the interface. This is not actually true. The bulk fully determines a "minimal" model of the gapless theory -- however, there is nothing to prevent one from having additional, unprotected modes at the interface. But furthermore, one can have a situation where a single gapped bulk can be bounded by more than one possible edge theory (this is known as stable equivalence, and is described in detail in a paper by Cano et al, PRB 89 115116 2014, among other places). I admit that there it is probably too much off the topic of the current paper to discuss this, but one does want to avoid saying things that are strictly incorrect.

A second issue of language is the use of the word "adiabatic transformation" (which usually refers to slow in time) and "transition between two phases" (which usually refers to changing the global hamiltonian). While most people will figure out what you mean, it is better to use precise language if possible.

Response to all reviewers and summary of changes

Summary of changes in the order as they appear in the revised manuscript (order as changes appear in the manuscript):

1. The title was changed to "Model States for a Class of Chiral Topological Order Interfaces". There and throughout the paper, we have replaced the ambiguous expression "variational ansatz" to "model state".
Indeed, and as pointed out by Referee 2, our ansatz, just like the celebrated Laughlin wavefunction, has no (continuous) variational parameters. Thus, the new terminology is less ambiguous about the nature of our wavefunction. We also followed the suggestion of Referee 3, making clear that our construction relies on a specific embedding and is applicable as such only to a class of chiral topological order interfaces.
2. In the introduction, we have included some additional references about the breakdown of the bulk-edge correspondence. These works go beyond the scope of the papers. But they are indeed worth mentioning to provide a more accurate picture, as pointed out by Referee 3.
3. A paragraph was added to section "Setup" in order to elaborate on the specific structure of the Hilbert space that allows us to write the model wavefunction. The embedding and the choice of basis in which we represent the vertex operators is made explicit, as suggested by both referees 1 and 2.
4. A figure was added to the section "Model Wavefunction" which both illustrate the ideas presented in the previously mentioned added paragraph, and makes clear how the MPS boundary state on the Laughlin side may be used to change the state at the interface. The rest of the section was slightly reorganized to extend the discussion about the construction of the ansatz – with Eq. 2 added as suggested by referee 3 – and to accommodate the new figure.
5. The supplementary materials now contain all the results about the consistency checks, convergence properties and truncation effects. The goal is to provide the full details of the numerical data needed to assert our claims. As pointed out by Referee 1, we did put a lot of efforts and invested weeks of numerical simulations in the careful characterization of the low-energy excitations of the systems or in the generalization of the computation to the fermionic case, more relevant for near term experimental applications.

Referee 1 is right in saying that the 14 pages of supplementary materials obscure the physics we want to put forward, especially since we only briefly mention these works in the main text. She or he says that "Much of the material that has been relegated to the supplementary information would have been completely appropriate for the paper". We will present these results in a separate publication, where these non-trivial computations will be discussed more thoroughly, complemented by a more detailed study of the fermionic case, which is more experimentally relevant (as discussed recently in arXiv:1810.06036).

6. We have fixed several typos and reformulated a few points according to the referees' suggestions.

Reply to the first referee

We thank the referee for recommending to publish our paper after her/his careful examination of both the manuscript and the supplementary materials and for her/his insightful comments. We acknowledge her/him stating that "the work presented in the paper and in the supplementary data is excellent". We greatly value her/his remarks and suggestions, and we have tried to address them all.

More specifically, we have clarified how the MPS boundary state on the Laughlin side may be used to dial the states at the interface by extending the discussion about the basis used for the CFT Hilbert space. We have also added a new figure (Fig. 1) to illustrate our point. The full auxiliary space, that of a two-component free boson, may be written as $\text{Aux}_H = \text{Aux}_L \otimes \text{Aux}_\perp$ (up to compactification conditions) where the Laughlin iMPS matrices act as the identity on Aux_\perp . States in Aux_\perp are labeled by a $U(1)$ -charge and a momentum which is left unchanged in the whole Laughlin bulk. It is thus transmitted to the interface where it meets the Halperin bulks and ceases to propagate. That is the knob we use to dial the states at the interface.

We have made clear in the caption of Fig. 3 and in the figures of the Supplementary Materials which curves were the prediction from the critical theory and which were the results of the fitting procedure. We have corrected some typos that the referee pointed out.

Most importantly, we are grateful for his comments about the long supplementary material and thought of the best way to find "the most appropriate form in which to present their findings". The referee was right in pointing out that the 14 pages of supplementary materials obscure the physics we want to put forward, especially since we only briefly mention these works in the main text and that "Much of the material that has been relegated to the supplementary information would have been completely appropriate for the paper". We have put a lot of effort in the characterization of the low-lying excitations, the comparison with finite size studies and the generalization to the fermionic case. We drastically changed the layout of the paper, removing from the supplementary materials the numerical studies that were only briefly mentioned or ignored in the main text. We believe these results may be more useful in a separate publication, where the non-trivial computations would be completed and emphasized, and the experimental consequences more detailed (as discussed recently in arXiv:1810.06036).

Reply to the second referee

We thank the referee for recommending to publish our paper in Nature Communications. Her/His concerns are about the clarity of the model state construction. We have changed two paragraphs and extended the discussion in order to answer to her/his questions.

More precisely, we added a paragraph in section "Setup" to show clearly why our ansatz crucially depends on the Laughlin $1/2$ auxiliary space being embedded in the Halperin 221 one (the CFT Hilbert space of a two-component free boson). Since all vertex operators are represented in the same basis, there is only one global gauge choice for both sets of matrices. Therefore, the ancillary matrix "gluing" together the two bulks is just the identity - or a completeness relation added between the two bulks. The global gauge, that we fix in our numerical simulations by picking a basis in which we represent the vertex operators, is one in which the previously mentioned embedding is transparent so that the Laughlin iMPS matrices are of the form $B_L \otimes \mathbf{I}$.

Furthermore, we have extended the discussion about the role of the boundary states mostly in the "Model Wavefunction" section and added Fig. 1 to illustrate this discussion. The referee asked how the momentum and charge sectors were coupled, and how the symmetries were implemented in the MPS. The matrices have a block structure with respect to the U(1)-charge and momentum of both component of the free boson, that we explicitly keep track of by selecting an eigen-basis of the Virasoro zero-th mode and of the bosonic currents zero-th modes. We studied these properties at length in another publication [Crépel et *al.*, Phys. Rev. B **97**, 165136], that we refer to for more details. Momentum and charge conservation at the interface are thus a consequence of the choice of basis (or the choice of global gauge) we make. We have written explicitly this property in the updated manuscript.

Referee 2 also mentions the general case for which one cannot rely on the structure we have just described. This is definitively something we would like to investigate in future works for it is a real challenge to find an ancillary matrix "gluing" together the two bulks in the most general setup. We do not know how to determine this ancillary matrix in general, but we thought of two approaches to tackle the problem. The first one is truly variational, and consists in minimizing the energy across the transition for different ancilla and will probably give results that we could compare to our Halperin-Laughlin ansatz as a starting point. The second method we could use is to perform a truncated conformal space analysis on an effective coupling Hamiltonian between the two edge modes of the bulks we would like to glue together (such as the one in Feverati et *al.* arXiv:hep-th/9803104 for instance). Remembering that the auxiliary spaces are nothing but the Hilbert spaces describing the edge modes of the system, the eigenvectors of such a coupling Hamiltonian may be recast in a matrix whose dimension fits the MPS bond dimension of the right and left bulks. Choosing the eigenvector minimizing the interface energy provides an ansatz whose relevance may be tested with the tools introduced in the article. Such a work is well beyond the scope of our article and will require enormous efforts and several technical advances.

Reply to the third referee

We are pleased to see the third referee's enthusiasm about our work and thank him for her/his detailed report. We also appreciate that she/he believes this work to be "an impressive piece of science".

We have addressed her/his remarks to be more cautious on a few terms we used, especially with the Bulk-Edge hypothesis - though she or he acknowledges that a full discussion would be beyond the scope of the paper. The works dealing with its breakdown and the study of its applicability however provide a much more accurate picture and we have included additional references on the subject to be complete.

We also think that the referee is right to propose to change the title since our construction strongly relies on the embedding of the Laughlin CFT space into the Halperin's one. The study indeed limits itself to "a Class of Chiral Topological Order Interfaces", as the new title suggests.

Finally, we would like to comment on her/his comment about the applicability of our method to other systems and about the generic case that the referee mentioned. We discuss in the article some other systems where our method may be applied and for which an embedding of one auxiliary space in another is known (see the reference in the article, for instance Grosfeld et *al.* Phys. Rev. Lett. **103**, 076803 or Bais et *al.* Phys. Rev. Lett. **102**, 220403). We agree that there are "cases where there nontrivial things happening at the interface, which still do not fit into the structure discussed in this work - or at least not in any obvious way" when there is no possible embedding of an auxiliary space into the other. In order to tackle those problems, and as explained in our answer to Referee 2, we plan to introduce an effective coupling of the edges and numerically diagonalize it to find the ancillary matrix which minimize the interface energy. The solution at large coupling may be inferred as a renormalization group fixed point of the CFT perturbed by the coupling between the chiral and anti-chiral edge modes. These studies, numerical truncated conformal space approach or perturbed CFT analysis, are highly challenging. They are highly relevant though (both from the pure CFT side, the topological order side and the experimental perspectives). Thus, we will pursue this effort in the following years.

REVIEWERS' COMMENTS:

Reviewer #1 (Remarks to the Author):

Dear Authors and Dear Editors of Nature Communications,

First of all, please excuse me for the long delay in producing this review.

Earlier, I reviewed the manuscript "A variational approach to chiral topological order interfaces", by the same authors. I considered this to be excellent work and recommended it for publication, although I had some reservations about the form of the paper (particularly the balance between the paper and the supplementary material) and some minor requests for clarifications and corrections.

The current paper, "Model States for a Class of Chiral Topological Order Interfaces", represents what is essentially a subset of the content of the previous paper, but in a clearer way and with a much better balance, both in quantity and in content, between the paper itself and the supplementary material.

I can paraphrase from my earlier report that this paper represents an important advance in the description of systems which have spatial regions in different topologically ordered phases, with nontrivial low energy degrees of freedom on the edges between these regions. It introduces trial wave functions for an important class of examples of such systems which may be experimentally realized, for example in fractional quantum Hall systems, particularly in fractional quantum Hall states in graphene. A number of important consistency checks are done on these wave functions, showing that they reproduce the topological orders of the two constituent bulk states as well as the features of the interface which have been theoretically predicted from effective (but non-microscopic) models.

The current paper does not check the trial wave functions against numerically exact results for small systems, as this is now delegated to the companion paper, "Microscopic Study of the Halperin-Laughlin Interface through Matrix Product States".

I do not think this was necessarily the most natural split of the work, as it would be more usual to propose trial wave functions and present numerical verification at the same time where this is possible, but I do not object too strongly, because the current paper is more readable and balanced than the previous manuscript and the careful test against exact numerics can be easily found in the companion paper. One remark to be made is that the verification of the state discussed in this paper actually occurs in the supplement to the companion paper and it may be useful to indicate this explicitly in the reference in the current paper, assuming the companion paper is published in the current form. All in all, I still think the work presented in this paper and in the supplementary data is excellent and by itself of enough importance for a publication in Nature Communications and I recommend it highly for publication.

I do have some very minor remarks which the authors may want to take into account and also noticed some typos which may escape a spell checker. I list these below.

1. First of all I would like to thank the authors for their more explicit account of the split of the Hilbert space which allows for the separation between Laughlin like and other degrees of freedom, and for a simple description of "knob" that dials the edge charge. This helped me :)

2. Pg. 2, col. 1. The reference to Fig. 2a should be to Fig. 1

3. Pg. 2 col. 2. "Hiding the contraction over the physical indices".

For clarity, especially for those readers not well versed in matrix product state methods, it might be good to be a little more explicit at least once in the paper.

4. Above equation (4) "electrons" should be "bosons"

5. Fig. 4 inset. The text in the caption appears to imply that the shaded area is the excess charge, but of course the left hand shaded area in the inset makes a negative contribution to the charge excess. Not sure if this is worth clarifying more.

6. Pg. 5 col 1 "In the later case" -> "In the latter case" Similarly for the other occurrence of "later" on pg. 2 of the auxiliary material ("The later point")

7. pg. 3 supplement "over a few magnetic length" -> lengths
"which is only entirely capture" -> captured

Reviewer #2 (Remarks to the Author):

The split version of the article makes it far more readable, the interface ansatz is now clear from the main text. While hidden by its modesty, the work is a numerical tour de force - I am very impressed with how nicely it works.

I have no substantial comments, but did want to confirm that for the comparison "We find extremely good agreement between the ED ground state and our MPS ansatz, with an overlap of 0.9989", you switch to the finite MPS ansatz presumably with vacuum boundary conditions at the edge?

One interesting direction I would hope you investigate in the future is Haldane's proposal that at a boundary between Hall fluids with different Hall viscosity, there is a universal, quantized dipole moment per unit length. I would guess this is not quite correct, and your ansatz would give an interesting way to probe it (though it may be correct in the special case of zero-energy interfaces like the 221 - (2) you study here).

Reviewer 1 (Remarks to the Author):

« Dear Authors and Dear Editors of Nature Communications,

First of all, please excuse me for the long delay in producing this review. Earlier, I reviewed the manuscript "A variational approach to chiral topological order interfaces", by the same authors. I considered this to be excellent work and recommended it for publication, although I had some reservations about the form of the paper (particularly the balance between the paper and the supplementary material) and some minor requests for clarifications and corrections.

The current paper, "Model States for a Class of Chiral Topological Order Interfaces", represents what is essentially a subset of the content of the previous paper, but in a clearer way and with a much better balance, both in quantity and in content, between the paper itself and the supplementary material.

I can paraphrase from my earlier report that this paper represents an important advance in the description of systems which have spatial regions in different topologically ordered phases, with nontrivial low energy degrees of freedom on the edges between these regions. It introduces trial wave functions for an important class of examples of such systems which may be experimentally realized, for example in fractional quantum Hall systems, particularly in fractional quantum Hall states in graphene. A number of important consistency checks are done on these wave functions, showing that they reproduce the topological orders of the two constituent bulk states as well as the features of the interface which have been theoretically predicted from effective (but non-microscopic) models.

The current paper does not check the trial wave functions against numerically exact results for small systems, as this is now delegated to the companion paper, "Microscopic Study of the Halperin-Laughlin Interface through Matrix Product States".

I do not think this was necessarily the most natural split of the work, as it would be more usual to propose trial wave functions and present numerical verification at the same time where this is possible, but I do not object too strongly, because the current paper is more readable and balanced than the previous manuscript and the careful test against exact numerics can be easily found in the companion paper. One remark to be made is that the verification of the state discussed in this paper actually occurs in the supplement to the companion paper and it may be useful to indicate this explicitly in the reference in the current paper, assuming the companion paper is published in the current form. All in all, I still think the work presented in this paper and in the supplementary data is excellent and by itself of enough importance for a publication in Nature Communications and I recommend it highly for publication.

I do have some very minor remarks which the authors may want to take into account and also noticed some typos which may escape a spell checker. I list these below.

1. First of all I would like to thank the authors for their more explicit account of the split of the Hilbert space which allows for the separation between Laughlin like and other degrees of freedom, and for a simple description of "knob" that dials the edge charge. This helped me :)

2. Pg. 2, col. 1. The reference to Fig. 2a should be to Fig. 1

3. Pg. 2 col. 2. "Hiding the contraction over the physical indices". For clarity, especially for those readers not well versed in matrix product state methods, it might be good to be a little more explicit at least once in the paper.

4. Above equation (4) "electrons" should be "bosons"

5. Fig. 4 inset. The text in the caption appears to imply that the shaded area is the excess charge, but of course the left hand shaded area in the inset makes a negative contribution to the charge excess. Not sure if this is worth clarifying more.

6. Pg. 5 col 1 "In the later case" -> "In the latter case" Similarly for the other occurrence of "later" on pg. 2 of the auxiliary material ("The later point")

7. pg. 3 supplement "over a few magnetic length" -> lengths "which is only entirely capture" -> captured

»

Reply to Reviewer 1:

We are pleased that the current manuscript is seen as "clearer" and "with a much better balance, both in quantity and in content, between the paper itself and the supplementary material" by Reviewer 1. We thank her/him again for encouraging us to find this more appropriate form to present our finding in his previous report. Moreover, we are deeply grateful for the three detailed and insightful reports she/he issued during the referral process.

We have addressed her/his remarks as detailed below:

- We introduced the many-body occupation basis to make clear the role of the physical indices in Eq. (2) and in the text below.
- We have corrected the typos she/he noticed and corrected the reference to Fig. 1 in the Results section.

Reviewer 2 (Remarks to the Author):

« The split version of the article makes it far more readable, the interface ansatz is now clear from the main text. While hidden by its modesty, the work is a numerical tour de force - I am very impressed with how nicely it works.

I have no substantial comments, but did want to confirm that for the comparison "We find extremely good agreement between the ED ground state and our MPS ansatz, with an overlap of 0.9989", you switch to the finite MPS ansatz presumably with vacuum boundary conditions at the edge?

One interesting direction I would hope you investigate in the future is Haldane's proposal that at a boundary between Hall fluids with different Hall viscosity, there is a universal, quantized dipole moment per unit length. I would guess this is not quite correct, and your ansatz would give an interesting way to probe it (though it may be correct in the special case of zero-energy interfaces like the 221 - (2) you study here). »

Reply to Reviewer 2:

We thank the referee for recommending to publish our paper in Nature Communications. We also acknowledge her/him stating that the article is now "far more readable". We thank him for suggesting a potential application of our method to one of Haldane's potential and we will investigate more thoroughly the feasibility of such a numerical study.

We also confirm, and written explicitly in the article, that the overlap with ED ground states required to switch to a finite MPS ansatz with vacuum boundary conditions at the edge.